# *"It reminds me and motivates me"*: Human-centered design and implementation of an interactive, SMS-based digital intervention to improve early retention on antiretroviral therapy: Usability and acceptability among new initiates in a high-volume, public clinic in Malawi

Jacqueline Huwa[1]*, Hannock Tweya[2], Maryanne Mureithi[3], Christine Kiruthu-Kamamia[1], Femi Oni[3], Joseph Chintedza[1], Geldert Chiwaya[1], Evelyn Waweru[3], Aubrey Kudzala[1], Beatrice Wasunna[3], Dumisani Ndhlovu[1], Pachawo Bisani[1], Caryl Feldacker[4,5]

1 Lighthouse Trust, Lilongwe, Malawi, 2 International Training and Education Center for Health (I-TECH), Lilongwe, Malawi, 3 Medic mobile, Nairobi, Kenya, 4 Department of Global Health, University of Washington, Seattle, WA, United States of America, 5 International Training and Education Center for Health (I-TECH), Seattle, WA, United States of America

* jhuwa@lighthouse.org.mw

## Abstract

### Background

Early retention of people living with HIV (PLHIV) in antiretroviral therapy (ART) programs is critical to improve individual clinical outcomes and viral load suppression. Although many mobile health (mHealth) interventions aim to improve retention in care, there is still lack of evidence on mHealth success or failure, including from patient's perspectives. We describe the human-centered design (HCD) process and assess patient usability and acceptability of a two-way texting (2wT) intervention to improve early retention among new ART initiates at Lighthouse Trust clinic in Lilongwe, Malawi.

### Methods

An iterative HCD approach focused on patient and provider users' needs, incorporating feedback from multidisciplinary teams to adapt 2wT for the local, public clinic context. We present mixed-methods usability and acceptability results from 100 participants, 50 at 3-months and 50 at 6-months, post 2wT enrollment, and observations of these same patients completing core tasks of the 2wT system.

### Results

Among the 100 usability respondents, 95% were satisfied with visit reminders, and 88% would recommend reminders and motivational messages to friends; however, 17% were

**Data Availability Statement:** All relevant data are within the manuscript and its Supporting information files. Qualitative data is also now attached as a supplementary file.

**Funding:** Research reported in this publication was supported by the Fogarty International Center of the National Institutes of Health (NIH) under Award Number R21TW010583 (CF). The content is solely the responsibility of the authors and does not necessarily represent the official views of the National Institutes of Health. The funders had no role in study design, data collection and analysis, decision to publish, or preparation of the manuscript.

**Competing interests:** The authors have declared that no competing interest exist.

worried about confidentiality. In observation of participant task completion, 94% were able to successfully confirm visit attendance and 73% could request appointment date change. More participants in 4–6 months group completed tasks correctly compared to 1–3 months group, although not significantly different (78% vs. 66%, p = 0.181). Qualitative results were overwhelmingly positive, but patients did note confusion with transfer reporting and concern that 2wT would not reach patients without mobile phones or with lower literacy.

## Conclusion

The 2wT app for early ART retention appears highly usable and acceptable, hopefully creating a solid foundation for lifelong engagement in care. The HCD approach put the local team central in this process, ensuring that both patients' and Lighthouse's priorities, policies, and practices were forefront in 2wT optimization, raising the likelihood of 2wT success in other routine program contexts.

## Introduction

Poor retention of people living with HIV (PLHIV) on antiretroviral therapy (ART) negatively affects individual clinical outcomes and viral load suppression [1]. In sub-Saharan Africa (SSA), in particular, suboptimal retention results in missed targets for critical UNAIDS 95-95-95 targets by 2030 [2]. A systematic review of ART retention in SSA estimated 36-month retention at 65% [3]. Early retention, within the first 6 months on ART, can be the most challenging, resulting in concentrated treatment disruptions within the first year on ART [4]. Factors influencing retention on ART are complex and dynamic, including individual (e.g., poverty, poor access to care, work or familial obligations), institutional (e.g. healthcare workers (HCW) shortages) and macro-level factors (e.g., COVID-19; global funding priorities) [1, 5–7]. However, once patients miss visits or have treatment interruptions, early retention efforts are most likely to successfully return patients to care [8].

Digital or mobile health (mHealth) interventions may help reduce retention gaps [9]. One-way "blast" educational messages or reminders (pre-defined, no response) modesty improve retention and adherence [10, 11]. Adding interactive components, such as two-way texting, fosters direct patient to HCW communication and appears more effective at increasing engagement in care [12–14]. Still, many retention apps fail [15–18]. Across much of SSA, formidable technology, infrastructure, and connectivity challenges continue to stymie mHealth [19, 20]. Additionally, potentially increased workload or required levels of digital literacy among HCWs may limit adoption and potential scale-up [21–23]. Policy, governance structures, and changing standards present additional challenges that require additional stakeholder involvement from inception [24]. Lastly, the dearth of rigorous mHealth intervention evaluation reduces the evidence base to inform improvements [25]. Nonetheless, mHealth shows immense promise to support patients, providers, and programs. Digital health innovations must respond with clearer evidence of success or failure on which to build [26].

To accurately reflect the value and fit of innovation to a specific programmatic need, mHealth programs should prioritize assessment of patient usability and acceptability. According to the ISO/IEC 9126–1, *usability* involves understandability, learnability, operability and attractiveness—or how an intended user correctly or incorrectly interacts with a digital innovation [27]. Usability is often measured quantitatively. Acceptability is more focused on individual beliefs and practices or their attitude towards using the App and may be assessed qualitatively by

asking users how they perceive the App to meet their needs and preferences [28]. Usability and acceptability are both essential to designing effective mHealth interventions that may scale successfully [29]. To further amplify diverse stakeholder engagement and to ensure that the App matches diverse user demands, incorporation of elements of human-centered design (HCD)—such as iterative user feedback—can also benefit mHealth usability and acceptability [30–33].

In 2019, University of Washington's International Training and Education Center for Health (I-TECH) and technology partner, Medic, pioneered a two-way, text-based (2wT) approach for direct provider-to-patient communication for a randomized control trial (RCT) leveraging the open-source Community Health Toolkit (CHT) (https://communityhealthtoolkit.org/) and an HCD approach. The rigorous RCT demonstrated that interactive, hybrid, 2wT between male circumcision (MC) providers and patients during the 14-day post-operative period was as safe as in-person follow-up while significantly reducing provider workload [34]. Patients and providers rated the 2wT system highly for usability and acceptability [35], and 2wT lowered costs while improving post-operative care [36]. 2wT for MC success was likely due to adherence to many of the core characteristics of mHealth excellence, including prioritizing patient accessibility and acceptance, low technology costs, locally-led adaptation, strong stakeholder collaboration, and government partnership for sustained impact [37, 38]. 2wT for MC adapted successfully from RCT to routine settings [39], reaching over 22,000 men in post-MC care by mid-2022.

Building on the strength of the open-source, CHT-based 2wT and its HCD approach, I-TECH and Lighthouse Trust, the largest public provider of ART in Lilongwe, Malawi, partnered with Medic to adapt the 2wT system to improve early ART retention. Lighthouse Trust is a World Health Organization (WHO) recognized centre of excellence, a public Ministry of Health (MoH) provider, and a leading organization in Malawi's HIV/AIDS response [40]. Lighthouse has five centers of excellence across the country, including the Martin Preuss Clinic (MPC). MPC is the largest facility with over 25,000 patients on ART. All diagnosed HIV-positive individuals who report to clinic reception are registered in the point of care (POC) electronic medical record system (EMRS). Since 2006, the Lighthouse Trust implemented a Back-to-Care (B2C) program to increase long-term ART retention. The B2C program identifies and traces patients who miss clinic appointment by at least 14 days and supports them to continue ART. Although B2C retention efforts at Lighthouse are robust [41, 42] human and financial resource constraints stretch scare resources, leaving B2C service gaps. In public ART clinics in Malawi like Lighthouse, digital health interventions like 2wT could improve early retention while simultaneously reducing patient tracing workload and costs.

Therefore, in 2021, Lighthouse, I-TECH, and Medic developed a 2wT-based intervention to improve ART retention and initiated a quasi-experimental study to determine 12-month retention impact by comparing 500 2wT participants to a historical cohort of 500 new ART initiates at Lighthouse's MPC Clinic in Lilongwe, Malawi. In this process-oriented paper, we describe the HCD process for the 2wT for early ART retention app. We present mixed-methods results on patient usability and acceptability with 100 early participants, 50 at 3-months and 50 at 6-months post 2wT enrollment, and report observations of participants interacting with the system. We hypothesized that using an HCD approach and intensive local participation in 2wT optimization would result in high usability and acceptability of 2wT from the patient perspective.

## Methods

### Study setting

At Lighthouse, patients are initiated on ART following the test and treat strategy, which includes immediate ART initiation for all patients tested HIV positive regardless of clinical or

immunological stage [43]. All new initiates receive augmented adherence and HIV disclosure counselling as part of ART initiation. New ART patient visits are scheduled monthly during the first six months on ART and then every three- or six-months if the patient is stable and adherent to ART. Ideally, the B2C program identifies and traces patients who miss clinic appointment by at least 14 days; however, delays in tracing patients are not uncommon, and treatment interruptions may continue beyond 2 weeks. At 60 days, patients failing to return are considered lost to follow-up (LTFU). All ART care, including B2C, is provided free to patients as mandated by MoH.

## Study population and recruitment

In the overall, parent quasi-experimental, pre-post study of retention outcomes, eligibility criteria for opt-in 2wT enrolment are: 1) Patients who initiated ART within 6 months; 2) ≥18 years; 3) possession of own phone at enrolment; 4) willing to receive/send SMS; 5) informed consent; and 6) receives confirmed 2wT enrolment text to verify enrolment number. Those without cell phones and who did not wish to participate were excluded from the study and received routine B2C retention support.

Information about 2wT was disseminated during routine patient care as part of ART initiation counseling. New ART initiates who completed B2C locator forms and had mobile phones with them were requested to enroll in 2wT study. Interested patients were then referred to the 2wT site study coordinator for informed consent and enrollment. Those who successfully received a confirmation text were enrolled. Participants were given a thorough overview of the SMS system, including required replies and instructions on how to confirm a clinic appointment, report a transfer out, or reschedule a clinic visit. A total of 500 participants, including 50 from a 2wT system pilot, were enrolled in 2wT group and 500 in the retrospective comparison group for the parent study.

The 100 2wT participants for the usability and acceptability assessment were recruited between August 2021 and January 2022 and selected via a purposeful, strategic sampling approach, enrolling the first 50 participants to reach 6- and 3-months, post 2wT launch, without overlap participants between groups.

## 2wT system development

**Technology background..**   Since 2008, Medic equipped more than 41,000 health workers to support more than 80 million care activities across 15 countries [44]. Medic is the steward of the CHT open source project that currently supports almost 60 apps across 23 unique workflows [44, 45], including 2wT-based apps in Malawi, South Africa, and Zimbabwe. Apps built using the CHT work on basic phones, smartphones, tablets, and computers with or without internet connectivity, in any language [46] and are tailored to meet key digital health characteristics recommended by the WHO [33]. The core areas of functionality used by the Malawi 2wT system include peer-to-peer and automated interactive messaging, task management features (unread message; message not delivered), longitudinal patient records, routine syncing, and reports for routine monitoring (e.g. patient response rates, referrals, etc.).

**2wT human-centered design..**   The Medic-led HCD approach for 2wT for early ART retention [30] focused on patient and provider users' needs, working in collaborative, multidisciplinary teams (Fig 1). Before 2wT system adaptation, system developers, HCWs, monitoring and evaluation (M&E) teams, B2C staff and patients discussed and specified the context use. A brief formative research was conducted to inform adaptation and implementation. Then, a pilot of 50 ART patients solicited end-user experience in message context, optimal message delivery timing and language preferences (Chichewa or English) with the aim to

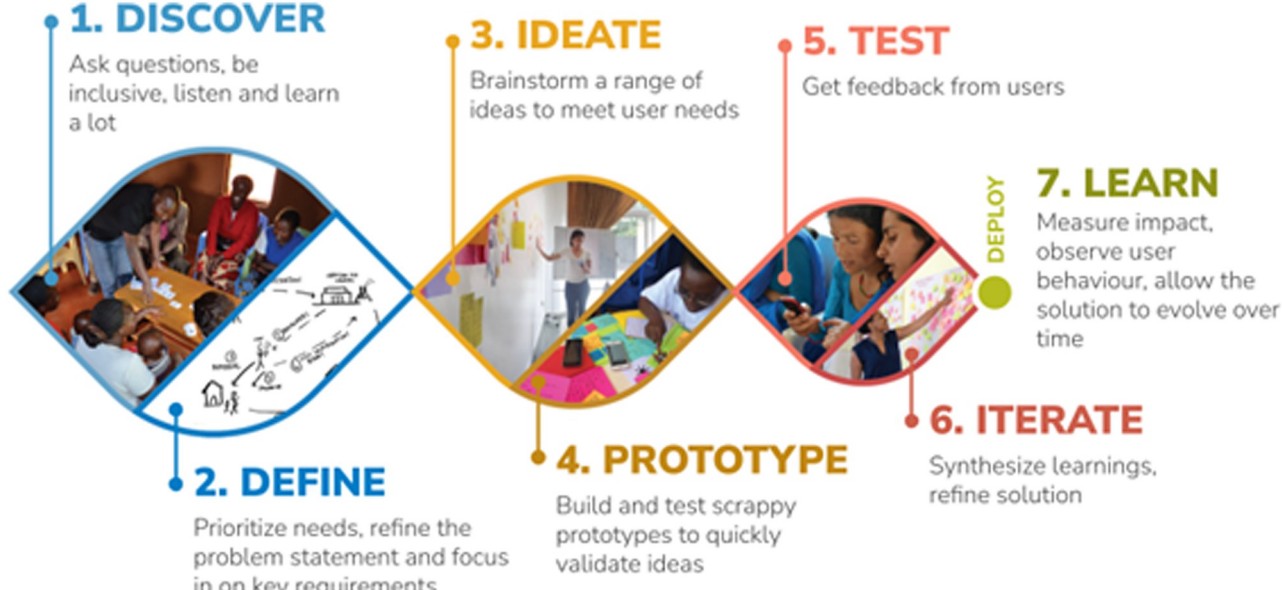

**Fig 1. Medic's human-centered design (HCD) approach.** Medic's HCD approach was utilized for designing, developing and optimizing 2wT for MC post-operative care [30, 47]. Republished from [47] under a CC BY license, with permission from B. Wasunna, original copyright, Medic 2020.

launch a patient-centered, usable, and acceptable app. Formative message research and pilot testing results also ensured participants' privacy and confidentiality were protected, making changes to messaging protocols to improve patient protections. Prior to updating the system messages, text messages were reviewed by expert patients (experienced ART patient mentors) for clarity and understandability. After system deployment, weekly technical feedback meetings were conducted to provide a platform to incorporate the voices and experiences of patient users in further refinement processes.

**System components and functionalities..** The 2wT system (Fig 2) is a hybrid automated and manual texting interaction between patients and healthcare workers. In the 2wT system, weekly motivational messages are sent without a response request. Individually tailored appointment reminders are sent in anticipation of a visit with a response requested. Multiple responses and free text are accepted at any time. Messages are sent either in English or Chichewa according to patients' preferences. Patients can send a message with any content at any time in any language. Generally, the 2wT officer responds to any SMS during routine clinic hours within 1–2 days of any message. Patients can opt out of motivational and/or visit reminder messages at any time.

**2wT intervention: Weekly motivational messages..** The 2wT system broadcasts weekly motivation messages with a new health-related message sent each week; example are shown in Table 1. The messages are sent from the same Lighthouse aggregator number and contain no information related to HIV or AIDS. Messages repeat after approximately 6 months. For weekly motivational messages, there is no response required nor requested from patients. Patients may respond if they choose.

**2wT intervention: Visit reminders and responses..** The 2wT system sends individualized clinic visit reminders with a response requested. Patients are asked to respond to the visit reminder messages with a single number: 1 = yes if confirming clinic attendance or 0 = no if they will not attend (Fig 2). A "yes" triggers, "See you soon!" and stops visit reminders for that specific visit. A "no" triggers automated, scripted texts to determine if someone has transferred

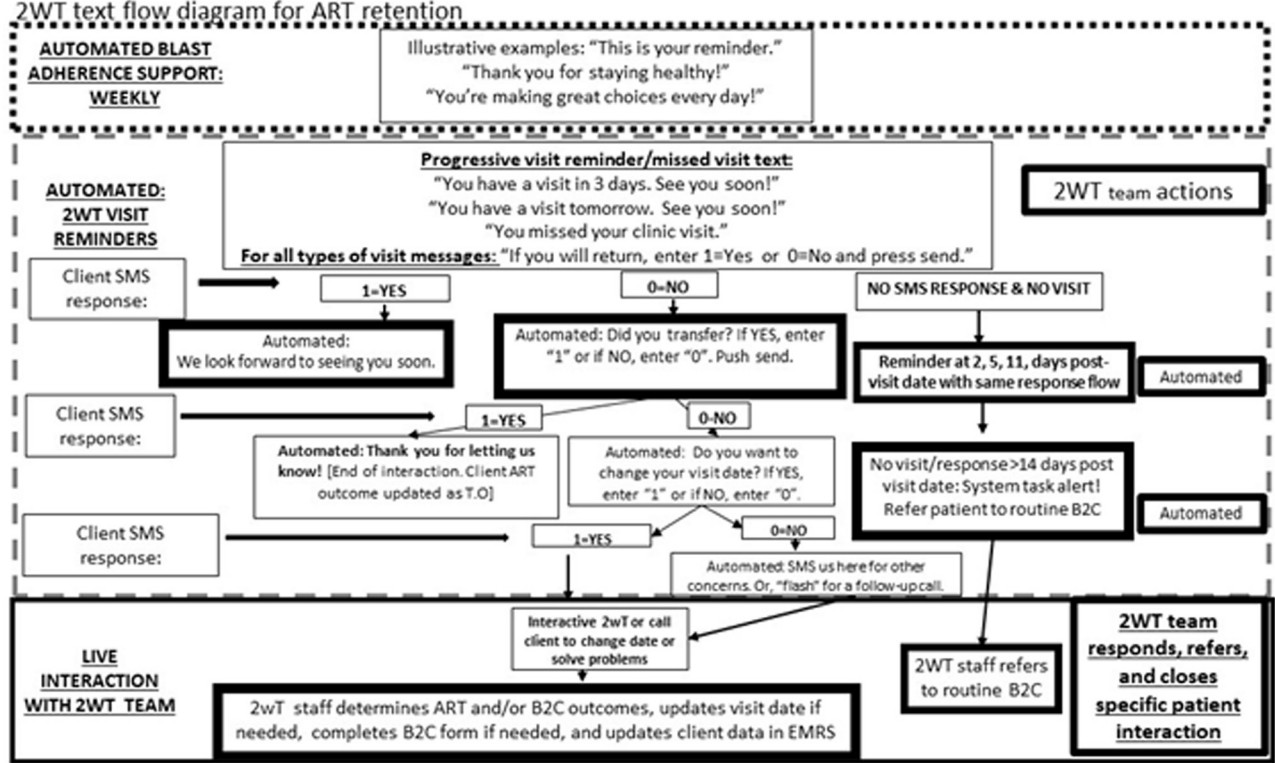

**Fig 2. 2wT flow diagram for ART retention.** This hybrid automated-interactive 2wT flow diagram illustrates typical pathways of client interaction with the system and 2wT officers.

**Table 1. A sample of motivational messages sent by the 2wT for ART retention.**

| Category | 2wT Messages |
|---|---|
| **Nutrition** | Keep making good decisions by eating healthy foods such as fruits, beans, and greens. |
| | Drink boiled or chlorinated water, 2 to 4 litres a day. |
| | Try cooking with a little less salt and oil to prevent high blood pressure (BP) and diabetes. |
| **Mental health** | Time with friends is a great way to reduce stress. Go chat with a friend! |
| | Laughter is good medicine. Let yourself laugh and smile! |
| | A little movement goes a long way for mental health. Take a walk. Be outside. Smile. |
| **Exercise, and General Health** | Stay active to keep your body healthy and strong to help prevent high blood pressure (BP), heart problems, diabetes, and stroke. |
| | Keep your home clean. It will help you have a healthy mind and body. |
| **COVID** | Get vaccinated for COVID-19 vaccine. It is free! The vaccine is safe for you, as well as your friends. |
| | Getting vaccinated for COVID-19 will prevent severe Covid-19 infection. |
| | COVID-19 is still a threat to our lives. Let us continue to wear mask properly, by covering our nose and mouth. |
| | COVID-19 is still a threat to our lives. Let us continue to gather with friends and family outside. |

or wants to change their visit date. The system also allows for free text interaction between 2wT staff and the patient to determine next steps. Patients can request an appointment date change, report transfers, or request a call. Patients receive automated missed visit alerts on days 2, 5, and 11 post appointment date when a clinic appointment is missed. Motivational and reminder visit messages are muted if or when specific ART outcomes are ascertained (stopping ART, transferring to other clinic and death). A 2wT officer prompt for B2C tracing is automatically triggered by the system when a patient misses a clinic appointment by 14 days.

### Data collection and management

A brief, semi-structured, interviewer-administered questionnaire (S1 File) contained 16 closed-ended questions to assess 2wT usefulness, acceptability, perceived benefits, and barriers. In open-ended questions, patients were asked for feedback on the intervention, express concerns and suggest improvements.

Participants were also observed completing key 2wT tasks to assess the effectiveness of 2wT education at enrollment and to help improve participant's adherence to 2wT prompts. Participants were shown four pictures of 2wT visit reminder messages and asked to indicate, using words or touching their phone, how they would: (a) confirm visit attendance, (b) transfer out to another clinic, (c) change appointment dates; (d) ask for help with something else (Fig 3). Results from assessment methods were combined to understand 2wT usability and acceptability from patient users.

### Statistical analysis

Quantitative data analysis was conducted using Stata software version 17.0 (S2 File). For questionnaires on acceptability, frequencies were used to explore closed-ended or partially open-ended responses (S3 File Short answers). Atlas.ti 8 was used for qualitative data from open-ended short answers. Qualitative data were entered and coded by one coder using a mix of

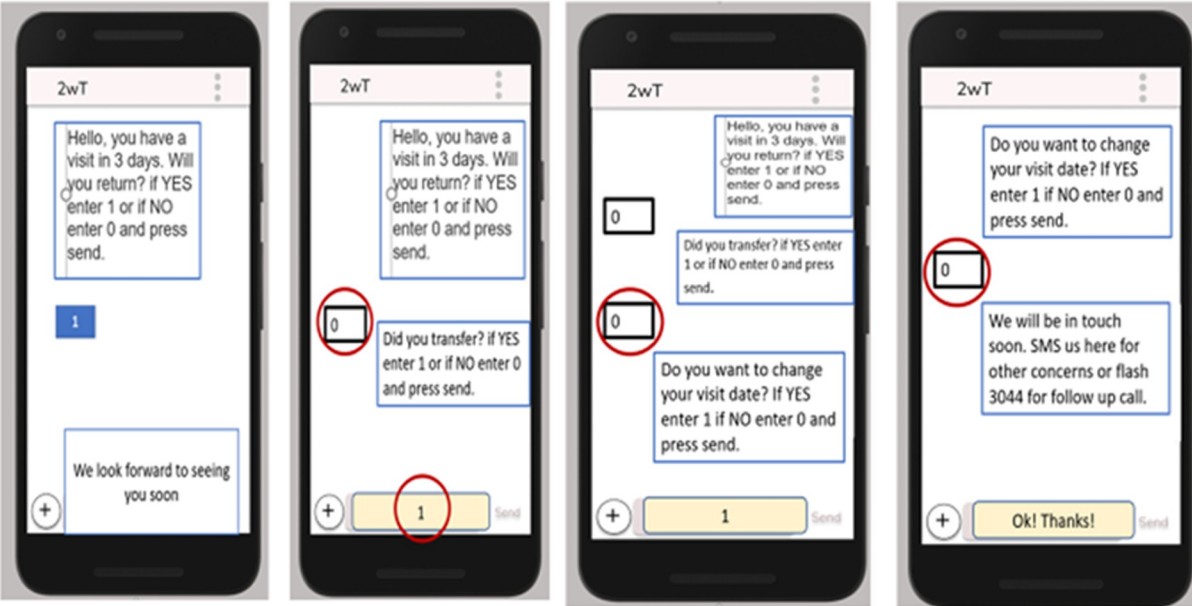

**Fig 3. Usability assessment of 2wT messaging on feature or smart phones.** Answer/education sheet showing clients how to respond to visit reminders, transfer reporting, visit date change requests, and requests for additional help.

deductive and inductive approaches guided by the questionnaire and thematic content. Initial thematic content was shared with the study team for feedback and revisions incorporated into final results.

## Ethics

The parent study, including the usability and acceptability components, was reviewed and approved by the University of Washington and the Malawi National Health Sciences Research Committee. All participants provided written informed consent.

## Results

### Demographic characteristics

One hundred 2wT participants were included in this sub-study (Table 2), including 28 from the pilot period. On average, participants were age 37.1 (SD 9.4) years; almost 40% of participants were under 35 years old. The majority (≥67%) of the participants started ART with WHO stages 1 or 2.

### 2wT app usability

Most participants were satisfied with visit reminders (95%) and the motivational messages (90%) (Table 3). Over 88% would recommend both. However, 9% did not understand the weekly motivational messages schedule, while over 25% expressed challenges responding correctly to reminders. There were also some confidentiality concerns: 17% were worried about breaches.

### 2wT task completion

All 100 participants completed four tasks designed to test their ability to respond to 2wT messages. Overall, 72% answered correctly to interactive message task: 94% visit attendance confirmation, 72% informing the facility of transferring to another clinic, 73% changing clinic

**Table 2. Characteristics of ART patients using 2wT for ART retention in Lilongwe, Malawi.**

|  | Questionnaire group (N = 100*) | |
|---|---|---|
|  | **N** | **%** |
| Sex |  |  |
| Male | 50 | 51% |
| Female | 48 | 49% |
| Age |  |  |
| 18–34 | 39 | 40% |
| 35–45 | 38 | 39% |
| 45+ | 21 | 21% |
| WHO stage at ART initiation |  |  |
| 1 or 2 | 64 | 66% |
| 3 | 22 | 23% |
| 4 | 11 | 11% |
| Time in 2wT intervention |  |  |
| 1–3 months | 50 | 50% |
| 4–6 months | 50 | 50% |

*2 participants were not correctly matched to baseline data and were omitted from the demographic table, only

**Table 3. 2wT usability at 3- and 6-month periods of participation duration.**

| Question | Time in 2 WT | | | | | |
|---|---|---|---|---|---|---|
| | 1–3 months | | 4–6 months | | Total | |
| | 50 | % | 50 | % | 100 | % |
| Were you satisfied with the visit reminder messages? | | | | | | |
| Yes | 47 | 94% | 48 | 96% | 95 | 95% |
| No | 3 | 6% | 2 | 4% | 5 | 5% |
| Were you satisfied with the motivational messages? | | | | | | |
| Yes | 42 | 84% | 48 | 96% | 90 | 90% |
| No | 6 | 12% | 2 | 4% | 8 | 8% |
| Don't know | 2 | 4% | 0 | 0% | 2 | 2% |
| Did you understand that an SMS would arrive on your phone each week (motivational message)? | | | | | | |
| Yes | 43 | 86% | 48 | 96% | 91 | 91% |
| No | 7 | 14% | 2 | 4% | 9 | 9% |
| Did you have challenges responding to visit reminders? | | | | | | |
| Yes | 14 | 28% | 12 | 24% | 26 | 26% |
| No | 35 | 70% | 37 | 74% | 72 | 72% |
| Don't know | 1 | 2% | 1 | 2% | 2 | 2% |
| Did you have challenges receiving text messages (SMS)? | | | | | | |
| Yes | 6 | 12% | 7 | 14% | 13 | 13% |
| No | 44 | 88% | 43 | 86% | 87 | 87% |
| Were you worried about privacy receiving these text messages (SMS)? | | | | | | |
| Yes | 6 | 12% | 11 | 22% | 17 | 17% |
| No | 44 | 88% | 39 | 78% | 83 | 83% |
| Would you recommend motivational messages to friends? | | | | | | |
| Yes | 44 | 88% | 46 | 92% | 90 | 90% |
| No | 5 | 10% | 3 | 6% | 8 | 8% |
| Don't know | 1 | 2% | 1 | 2% | 2 | 2% |
| Would you recommend visit reminder messages to friends? | | | | | | |
| Yes | 39 | 87% | 36 | 88% | 75 | 87% |
| No | 6 | 13% | 4 | 10% | 10 | 12% |
| Don't know | 0 | 0% | 1 | 2% | 1 | 1% |

appointment dates task 1, and 99% changing clinic appointment date task 2 (Table 4). Overall, more participants in the 4–6 months group completed the task correctly compared to those in the 1–3 months group, although not significantly different (78% vs. 66%, p = 0.181). Both groups had difficulties completing the transfer-out and appointment dates tasks.

## Qualitative responses on acceptability and usability

Of 100 respondents, 95 provided responses to the open-ended questions (S3 File). Most respondents provided positive feedback and support for the 2wT intervention, asking the team to "*continue sending us messages, most people forget their clinic appointment."* The vast majority, noted their appreciation for the visit reminders, like this patient:

> "This system is good and continue offering us this service. We got reminded when we forget the dates. I came on 26 January for consultation, but I didn't get drugs then when my days were coming close, I received the reminder message. I thought in my mind that I had already received the meds. Thank you for reminding me."

**Table 4. 2wT task completion.**

| | Time in 2 WT | | | | | |
| --- | --- | --- | --- | --- | --- | --- |
| | 1–3 months | | 4–6 months | | Total | |
| Total | 50 | | 50 | | 100 | |
| All tasks | 33 | 66% | 39 | 78% | 72 | 72% |
| Visit attendance confirmation | | | | | | |
| No | 4 | 8% | 2 | 4% | 6 | 6% |
| Yes | 46 | 92% | 48 | 96% | 94 | 94% |
| Transfer out reporting | | | | | | |
| No | 17 | 34% | 11 | 22% | 28 | 28% |
| Yes | 33 | 66% | 39 | 78% | 72 | 72% |
| Change appointment date | | | | | | |
| No | 16 | 32% | 11 | 22% | 27 | 27% |
| Yes | 34 | 68% | 39 | 78% | 73 | 73% |
| Request help on something else | | | | | | |
| No | 1 | 2% | 0 | 0% | 1 | 1% |
| Yes | 49 | 98% | 50 | 100% | 99 | 99% |

Participants overwhelmingly noted that they felt encouraged and motivated by the messages, appearing to perceive the messages as coming from a member of the Lighthouse support team. Respondents, like this person, asked the team to, *"continue sending messages as they are encouraging us to stay healthy. It reminds me and motivates me."* Another participant noted, "*when I see the messages, I do have the conscious to take the medicine.*" Patients felt cared for and supported by the Lighthouse team, noting feelings of comfort such as, "through these interactions, *we are not afraid as we know you will help us,*" and "*I am very thankful when I receive these messages.*"

Patients also liked the weekly SMS health-related message, as one patient reporting: "*I am thankful because the messages remind me of my clinic appointment and encourages me to live a healthy life."* 2wT was also received positively as a good alternative to reminder phone calls. One participant expressed that "*SMS visit reminder is better than through phone call because colleagues ask us why someone is calling me."*

There were several themes that suggest the need for improvement. A few patients noted confusion with how to stop messages while others noted delays in HCW responses to their visit change or support requests. One patient also requested changes to support lower literacy patients, remarking that, *"the messages are good for people who know how to read. You should also prepare something for those who cannot read."* Lastly, a few patients receive missed visit reminders when they attended their visit on time, leaving them frustrated. *"Stop sending messages reminding us that we have missed the visit when we already reported."*

Not all responses were in support of 2wT. A few patients noted that even without explicit mention of ART nor the clinic name, they still had concerns about disclosure. One patient noted that they were "*always afraid that someone might see the messages as my phone is always with neighbors. Someone who knows the number XXXX might realize that this message is from Lighthouse and might compromise with my privacy."*

## Discussion

Likely as a result of the HCD-based adaption of the 2wT model, we found 2wT to be highly usable and acceptable as an early retention intervention for patients at MPC's routine ART

program. Although some patients, including those who are older or illiterate, may prefer reminder calls [14], these results indicate that the vast majority appreciated both the visit and motivational messages and would recommend 2wT for their friends. However, a quarter of respondents had trouble responding correctly to messages and there were some who expressed concerns with confidentiality, namely people seeing their messages. Compared to those in the 1–3 months in 2wT group, slightly more participants in the 4–6 month group correctly completed the tasks, showing that patient understanding and confidence likely increase over time. Although confirming visit attendance was considered simple, almost 30% found it difficult to report a transfer out or reschedule a clinic visit respectively. These findings suggest several strengths and areas of improvement for 2wT.

Iterative, highly-participatory, HCD was central to App optimization. Routine B2C retention staff helped co-design, develop, test, and optimize 2wT through a series of formative input sessions that also included ART patients, monitoring and evaluation staff. Cycles of optimization strengthened app specifics including: message frequency; motivational message content; timing of delivery; and wording of the automated messages. Pre-testing was also critical: the pilot test conducted among 50 participants prior to the roll-out helped to identify system weaknesses, coding flaws, and bugs which were rectified before study implementation. The hybrid approach, blending automated motivation messages and individually tailored visit reminders with access to direct, interactive, healthcare worker-to-patient messaging, put a human face to the intervention. Although SMS was the primary form of communication, phone calls were made at patient request. This flexibility to respond to patients via SMS or phone also demonstrates prioritization of patient preferences. Patients believed and were assured that a member of the Lighthouse team was supporting them in the manner they preferred, likely contributing to 2wT satisfaction.

Multi-method formative assessment of usability also added rich data on which to improve the system for patient users. Quantitative, observational, and qualitative data each provided key information to ensure that 2wT was both easy to use and highly useful to new ART patients, resulting in a system that helped patients attend visits and stay motivated in care. Other research also used multiple methods to help ensure adaptation and optimization for the local context. In South Africa, an app for HIV self-testing showed that nearly all (98.7%) found the app easy to use and reported that 89.0% of participants could follow all the steps on the app without error [48]. However, usability does not always indicate usefulness. In another South African app for HIV self-testing, high usability (87.8%) did not lead to participants to log onto the app: only 22% self- reported results [49]. In Malawi, assessment of two electronic community case management Apps to support decision making for childhood illnesses found significant differences between the two apps in ease of use [50].

There are several areas for improvement in the 2wT system. First, participants received training on responding to the 2wT system messages at enrolment only, which likely contributed to observed differences in usability over time. Practice and repetition, complemented by refresher patient education (in-person or via posters) during subsequent clinic visits may be recommended as patients tend to understand the system over time. Second, similar to other studies [51–53], confidentiality concerns or fear of disclosure of HIV status using SMS for ART reminders, were reported by some participants. Several protections were implemented to reduce confidentiality concerns. 2wT message co-creation and pre-testing with patients attempted to ensure that both motivational and visit reminder messages were neutral and benign, reducing potential associations with HIV-related care or services. 2wT was also implemented only as opt-in: patients were consented after 2wT educational counseling that included a presentation with example messages. Those who self-reported phone sharing were also excluded. However, this may not have been enough as 17% of the participants reported that

they were afraid someone else would see the messages. Additional protections should be explored. Lastly, 2wT is limited only to new ART enrollees and those with mobile phones and is intended to contribute to, and not replace, the routine Back to Care retention activities. However, reducing the literacy level required for participation or considerations of how to include those with shared phones would improve access to this retention support.

## Limitations

As with all interviewer-led questionnaires, it is possible that patients were affected by the interviewer, a bias that could have influenced responses in a more positive or negative direction. Usability could change over time. One hundred patients were included in the usability assessment, but de-identified data linked to study-specific IDs resulted in errors linking usability results to patient demographics, reducing the sample size for demographics from 100 to 98. Lastly, SMS response rates, retention outcomes and cost data is forthcoming. It is expected that these results will shed additional light on whether high usability and acceptability will result in effectiveness of the 2wT approach on retention as expected.

## Conclusions

2wT for early ART retention at Lighthouse Trust was purposefully designed with patients at the forefront. As a result, it appears that the App was highly usable and acceptable to those who chose to enroll in the retention support intervention, perhaps setting them up for a successful start to lifelong engagement in care. Although 2wT is not for all patients in all settings, the 2wT intervention approach has several strengths that are worthy of consideration for other digital health collaborators working to improve early ART retention. The local team led this process, ensuring that needs, priorities, and practices from Lighthouse, and not from international collaborators, were centered in the decision-making [54]. 2wT was purposefully built using the open-source Community Health Toolkit (CHT), a global digital good. By leveraging an existing set of tools, and adapting from previous 2wT-based evidence [34], delays with App development were reduced while the potential for sustainability increased. Lastly, as 2wT is implemented in a routine, MoH ART services context, the HCD process reflected the realities of routine, and not research, settings, raising the likelihood of longer-term fidelity and potential scale-up of the 2wT intervention in other LMIC programs.

## Supporting information

**S1 File. Usability survey.** This interviewer-administered survey was employed to assess usability.
(CSV)

**S2 File. Dataset.** Usability dataset in CSV format.
(CSV)

**S3 File. Dataset.** Qualitative short-answer responses from usability survey (English).
(PDF)

**S4 File.**
(DOCX)

## Acknowledgments

We would like to acknowledge the study participants, the study team (Kondwani Masiye, Harrison Chirwa, Blessings Wandira, Daniel Mwakanema, Madalitso Chawanje, William Maziya,

Isaac Nyirenda), colleagues from medic mobile (Edwin Kagereki, Adinan Alhassan, Mourice Barasa), the MPC clinic, M&E, retention team, and MoH staff at Bwaila for their valued contribution to the study.

## Author Contributions

**Conceptualization:** Hannock Tweya, Caryl Feldacker.

**Data curation:** Hannock Tweya, Caryl Feldacker.

**Formal analysis:** Hannock Tweya, Evelyn Waweru, Caryl Feldacker.

**Funding acquisition:** Caryl Feldacker.

**Investigation:** Jacqueline Huwa, Hannock Tweya, Maryanne Mureithi, Joseph Chintedza, Geldert Chiwaya, Caryl Feldacker.

**Methodology:** Jacqueline Huwa, Hannock Tweya, Maryanne Mureithi, Caryl Feldacker.

**Project administration:** Jacqueline Huwa, Joseph Chintedza, Geldert Chiwaya, Aubrey Kudzala, Pachawo Bisani.

**Resources:** Caryl Feldacker.

**Software:** Maryanne Mureithi, Femi Oni, Beatrice Wasunna, Dumisani Ndhlovu.

**Supervision:** Jacqueline Huwa, Hannock Tweya, Christine Kiruthu-Kamamia, Femi Oni, Joseph Chintedza, Geldert Chiwaya, Aubrey Kudzala, Beatrice Wasunna, Dumisani Ndhlovu, Caryl Feldacker.

**Validation:** Hannock Tweya, Maryanne Mureithi, Femi Oni, Beatrice Wasunna, Dumisani Ndhlovu.

**Visualization:** Hannock Tweya, Maryanne Mureithi, Femi Oni, Beatrice Wasunna, Pachawo Bisani.

**Writing – original draft:** Jacqueline Huwa, Hannock Tweya, Caryl Feldacker.

**Writing – review & editing:** Jacqueline Huwa, Hannock Tweya, Maryanne Mureithi, Christine Kiruthu-Kamamia, Femi Oni, Joseph Chintedza, Geldert Chiwaya, Aubrey Kudzala, Beatrice Wasunna, Dumisani Ndhlovu, Pachawo Bisani, Caryl Feldacker.

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
