## [Decision Letter · Decision Letter 0]

6 Apr 2023

PONE-D-22-32318“It reminds me and motivates me”: Human-centered design and implementation of an SMS-based digital intervention to improve retention on antiretroviral therapy: usability and acceptability among new initiates in a high-volume, public clinic in MalawiPLOS ONE

Dear Dr. Feldacker,

Thank you for submitting your manuscript to PLOS ONE. After careful consideration, we feel that it has merit but does not fully meet PLOS ONE’s publication criteria as it currently stands. Therefore, we invite you to submit a revised version of the manuscript that addresses the points raised during the review process.

We look forward to receiving your revised manuscript.

Kind regards,

Lisa Suzanne Dulli, PhD

Academic Editor

PLOS ONE

Journal Requirements:

2. We note that Figures 1 and 3 in your submission contain copyrighted images. All PLOS content is published under the Creative Commons Attribution License (CC BY 4.0), which means that the manuscript, images, and Supporting Information files will be freely available online, and any third party is permitted to access, download, copy, distribute, and use these materials in any way, even commercially, with proper attribution. For more information, see our copyright guidelines: http://journals.plos.org/plosone/s/licenses-and-copyright.

     1. You may seek permission from the original copyright holder of Figures 1 and 3 to publish the content specifically under the CC BY 4.0 license.

3. Please include a complete copy of PLOS’ questionnaire on inclusivity in global research in your revised manuscript. Our policy for research in this area aims to improve transparency in the reporting of research performed outside of researchers’ own country or community. The policy applies to researchers who have travelled to a different country to conduct research, research with Indigenous populations or their lands, and research on cultural artefacts. The questionnaire can also be requested at the journal’s discretion for any other submissions, even if these conditions are not met.  Please find more information on the policy and a link to download a blank copy of the questionnaire here: https://journals.plos.org/plosone/s/best-practices-in-research-reporting. Please upload a completed version of your questionnaire as Supporting Information when you resubmit your manuscript.

Reviewers' comments:

Reviewer's Responses to Questions

**Comments to the Author**

1. Is the manuscript technically sound, and do the data support the conclusions?

Reviewer #1: Partly

Reviewer #2: Partly

2. Has the statistical analysis been performed appropriately and rigorously? 

Reviewer #1: N/A

Reviewer #2: No

3. Have the authors made all data underlying the findings in their manuscript fully available?

Reviewer #1: Yes

Reviewer #2: Yes

4. Is the manuscript presented in an intelligible fashion and written in standard English?

Reviewer #1: Yes

Reviewer #2: Yes

5. Review Comments to the Author

Reviewer #1: In this article, the authors report mixed quantitative and qualitative findings from a quasi-experimental study of a two-way texting (2wT) app for appointment compliance for HIV treatment in Malawi. The authors report good end-user satisfaction and task completion for their 2wT app.

Given the benefit to public health, the authors’ work has considerable importance to the broader scientific and public health communities. Overall the article is well researched and well written, aside from a small number of minor errors listed below.

I have two major concerns however that I believe the authors must address prior to acceptance of their article.

Major comments

1.

The authors hypothesized that “an HCD approach and intensive local participation in 2wT optimization would result in high usability and acceptability of 2wT from the patient perspective.” However, the authors’ study design was not set up to test a null hypothesis. As such, it’s unclear how much of the end users’ satisfaction is ascribable to improvements in the 2wT app itself. The results are at severe risk of confound — confirmation bias from participating and novelty from the use of a new technology, for example, are likely contributing some unknown amount to self-reported satisfaction and task completion metrics.

2.

The scope of the authors’ discussion needs to be expanded beyond their own product. What was discovered in this process that will guide other digital- and telehealth projects? What insights are specific to HIV patients or to users in Malawi, and what insights can be extrapolated more generally to other mHealth developers? Currently, the discussion focuses on interpreting the authors’ own findings and not to generalizing for the benefit of other researchers.

Minor comments

From the manuscript, the authors engaged in pre-testing: a pilot test was conducted among 50 participants prior to the roll-out helped to identify system weaknesses, coding flaws, and bugs which were rectified before study implementation. What process was followed in this stage, how did it affect the creation of the 2wT app, and what results/insights can be extended to the broader research community?

P.5 88

I would assume privacy would be a motivating factor for MC as well, and yet privacy was less of a reported concern than for the 2wT app. Could this be driven by cultural, educational, or technology literacy effects?

P.8 145—153

I’m not sure if there’s a CoI issue but this paragraph is written very… positively… for Medic.

p.9 163-164

“Prior to updating the system messages, text messages were reviewed by expert patients (experienced ART patient mentors) for clarity and understandability.” What is meant by system messages and text messages in this context? Can this be clarified?

p.16 243-244

What is different about task1 and task2 to drive a failure rate of either 1% (task2) or 27% (task1)?

Grammar, language

p.4 59 modesty should be modestly

p.7 130 should be ‘or’ not ‘and’

p.8 149 I doubt ‘any language’, this should be amended

p.16 253 If the parenthetical 65 refers to the percentage of people (i.e. 65%) then I would not call this the “vast majority”

p.7 129 Own *cell* phone presumably

Reviewer #2: I would like to commend the authors for choosing the topic of interventions seeking to keep patients engaged in care and treatment. As we get closer to HIV epidemic control, retention into care and treatment becomes a significant concern for program managers and policy makers. Innovations such as the ones piloted and studied by the authors are extremely relevant and the use of digital solutions, most appropriate. The use of a human centered design is commendable. After reviewing the paper, there are several points I would like to raise that explain my recommendation:

1. The design of the study:

The study only describes the design of the parent study (quasi-experimental, pre-post study of retention outcomes) but not that of the sub-study. It will be good to have a description of the design for the sub-study, basis of this paper. The study design would benefit from having a control group. It seems like the parent study enrolled 500 participants in 2wT had a control group of 500. For the sub-study on usability and acceptability including 100 participants, it could be argued that there are two interventions introduced: i) the 2wT technology and ii) an enhanced orientation/counselling of participants on the use of the technology. The orientation/counselling of participants to the use of the 2wT technology was pretty intensive and potentially has a positive effect on the usability and acceptability responses. It is not clear that this enhanced orientation in a study setting is replicable in a routine service delivery setting where eventually this technology is intended to be taken to scale. A control group where the technology is deployed in a routine, non-research setting would have been helpful.

2. Social desirability bias

The authors rightly identify the possibility of social desirability bias, yet they do not offer an indication of how this bias was minimized. The enhanced orientation/counselling described above may have played a role. Asymmetry of information between health care providers and research staff as well as power differential in settings like Malawi health system are important dynamics to consider. Such bias undermines the validity of the qualitative results.

3. Sample and sample size:

A sample was drawn purposefully and strategically; it would be good to describe the purpose and strategy that guided the sampling. In addition, among the 100 participants recruited, 28 were from a pilot. It is not clear how having participated in a pilot influences the participants familiarity of the technology and therefore, likelihood to give favorable answers on the usability and acceptability.

6. PLOS authors have the option to publish the peer review history of their article (what does this mean?). If published, this will include your full peer review and any attached files.

Reviewer #1: No

Reviewer #2: No

---

## [Author Response · Author response to Decision Letter 0]

6 Jun 2023

13th May 2023

Dear Editor,

Thank you for facilitating the review of our manuscript submission. We would like to thank the reviewers for the constructive and detailed review, which we believe helps to improve our paper. We provide the following responses to the issues raised:

Reviewer #1

Major comments

1. The authors hypothesized that “an HCD approach and intensive local participation in 2wT optimization would result in high usability and acceptability of 2wT from the patient perspective.” However, the authors’ study design was not set up to test a null hypothesis. As such, it’s unclear how much of the end users’ satisfaction is ascribable to improvements in the 2wT app itself. The results are at severe risk of confound — confirmation bias from participating and novelty from the use of a new technology, for example, are likely contributing some unknown amount to self-reported satisfaction and task completion metrics.

Response: Thank you for this helpful comment. We recognize that using the word, hypothesis, was misleading. In this paper we describe the user centered design process for the 2wT app for ART retention. We did not test a null hypothesis. We removed the word, “hypothesis” and revised the language of the introduction to be clearer. 

This is a new app for retention. Before this intervention, expert clients sometimes texted there was only call and visit based retention. However, SMS is not a new technology. Throughout the descriptive paper, we note the HCD app development with high healthcare worker and PLHIV inputs. This is in complement to our submitted paper on engaging PLHIV and HCWs in the app design itself: 

 Mureithi, M., et al., Centering healthcare workers in developing digital health interventions: usability and acceptability of a two-way texting retention intervention in a public HIV clinic in Lilongwe, Malawi. medRxiv, 2023: p. 2023.01.09.23284326.

Although it is unknown whether desire to please the interviewer would lead to improved task completion. We noted the potential confounding in the limitations, “As with all interviewer-led questionnaires, it is possible that patients wanted to please the interviewer, perform better, or affirm the care providers, a response bias that could have led to greater positive app feedback. However, we engaged a well-trained, highly experienced qualitative interviewer who was not involved in the 2wT program and participants were assured of total confidentiality, likely reducing this bias. 

2. 

The scope of the authors’ discussion needs to be expanded beyond their own product. What was discovered in this process that will guide other digital- and telehealth projects? What insights are specific to HIV patients or to users in Malawi, and what insights can be extrapolated more generally to other mHealth developers? Currently, the discussion focuses on interpreting the authors’ own findings and not to generalizing for the benefit of other researchers. 

Response: This has been addressed. See page 20.338-344. Throughout the discussion, we note how our work could inform the work of others and bring in additional references to other work in the larger field. We added several ideas and a new paragraph. However, with the narrow usability and acceptability focus of this paper, we maintained fidelity to remain in scope. 

Minor comments

From the manuscript, the authors engaged in pre-testing: a pilot test was conducted among 50 participants prior to the roll-out helped to identify system weaknesses, coding flaws, and bugs which were rectified before study implementation. What process was followed in this stage, how did it affect the creation of the 2wT app, and what results/insights can be extended to the broader research community?

Response: A sister paper on the healthcare worker perspectives on the HCD process is under review with additional detail on this pre-pilot and post-pilot adaptation process (Mureithi, M., et al., Centering healthcare workers in developing digital health interventions: usability and acceptability of a two-way texting retention intervention in a public HIV clinic in Lilongwe, Malawi. medRxiv, 2023: p. 2023.01.09.23284326). However, by the pilot stage, and after the intensive HCD process, there were very few additional adaptations made. We clarified this in the HCD process paragraph. 

P.5 88 

I would assume privacy would be a motivating factor for MC as well, and yet privacy was less of a reported concern than for the 2wT app. Could this be driven by cultural, educational, or technology literacy effects? 

Response: Privacy concerns were indeed less reported for male circumcision (Feldacker et.al 2020) than for the 2wT for ART retention app, where 17% expressed privacy concerns. It appears HIV/ART is a more sensitive subject in our African society, compared to MC. While MC is culturally accepted in most societies, with the well-known benefit of reducing HIV acquisition, HIV is mostly associated with infidelity, although not everyone gets HIV through unprotected sex. As such, it is understandable that some participants would be worried receiving the messages for 2wT app. The study recruited participants who reported owning a personal mobile phone, but the fact that privacy was a matter of concern to some participants confirms that phone sharing cannot be completely excluded. The 2wT for ART retention app did not include any words or messages that would cause suspicion if seen by a third party, i.e. words like HIV, ART, clinic, Lighthouse etc.) 

P.8 145—153 

I’m not sure if there’s a CoI (conflict of interest) issue but this paragraph is written very… positively… for Medic.

Response: UW, Lighthouse and Medic co-designed the 2wT app which is open-source – not associated with any user fees nor application download costs. It is true that authors, like us, stand to gain recognition for our work as we publish. We take much pride in our work, reporting the positive and negative findings. We do note where this app tends to underperform and needs additional improvement. There is no product or personal gain from the 2wT app that would cause a conflict of interest in this context. Medic collaborators did contribute to this paper, but we feel secure in noting that CoI are not interfering with the statements and positions expressed in this paper. 

p.9 163-164 

“Prior to updating the system messages, text messages were reviewed by expert patients (experienced ART patient mentors) for clarity and understandability.” What is meant by system messages and text messages in this context? Can this be clarified?

Response: Text messages were first reviewed by expert patients before being updated in the system. We revised this message as follows: “Prior to finalizing automated text messages in the system (both motivational and visit-specific), both English and Chichewa text messages were reviewed by expert clients (experienced ART client mentors) for clarity of language and understandability in both English and Chichewa.”

p.16 243-244 

What is different about task1 and task2 to drive a failure rate of either 1% (task2) or 27% (task1)?

Response: Thank you for noting this. It is a mistake. It is now rewritten, “All 100 participants completed four tasks designed to test their ability to respond to 2wT messages. Overall, 72% responded correctly to all four tasks. For correct responses to specific tasks, 94% confirmed visit attendance, 72% confirmed informing the facility of transferring to another clinic, 73% confirmed a clinic appointment date change, and 99% confirmed ability to request more help (Table 4).

 Grammar, language

p.4 59 modesty should be modestly

p.7 130 should be ‘or’ not ‘and’

p.8 149 I doubt ‘any language’, this should be amended

p.16 253 If the parenthetical 65 refers to the percentage of people (i.e. 65%) then I would not call this the “vast majority”

p.7 129 Own *cell* phone presumably

Response: These have all been addressed 

Reviewer #2

Reviewer #2: I would like to commend the authors for choosing the topic of interventions seeking to keep patients engaged in care and treatment. As we get closer to HIV epidemic control, retention into care and treatment becomes a significant concern for program managers and policy makers. Innovations such as the ones piloted and studied by the authors are extremely relevant and the use of digital solutions, most appropriate. The use of a human centered design is commendable. After reviewing the paper, there are several points I would like to raise that explain my recommendation:

Response: Thank you for these kind comments. 

1. The design of the study:

The study only describes the design of the parent study (quasi-experimental, pre-post study of retention outcomes) but not that of the sub-study. It will be good to have a description of the design for the sub-study, basis of this paper. The study design would benefit from having a control group. It seems like the parent study enrolled 500 participants in 2wT had a control group of 500. For the sub-study on usability and acceptability including 100 participants, it could be argued that there are two interventions introduced: i) the 2wT technology and ii) an enhanced orientation/counselling of participants on the use of the technology. The orientation/counselling of participants to the use of the 2wT technology was pretty intensive and potentially has a positive effect on the usability and acceptability responses. It is not clear that this enhanced orientation in a study setting is replicable in a routine service delivery setting where eventually this technology is intended to be taken to scale. A control group where the technology is deployed in a routine, non-research setting would have been helpful.

Response: Thank you for these helpful comments. We agree that there is a technology intervention and a technology education or orientation component. The educational component is not directly assessed. However, as part of the usability, we did design complementary training and educational materials, including using the figure 3 for improved 2wT enrollment confirmation. While we cannot change the design of the usability assessment, we did add several important sentences to the paper. 

In the introduction of the paper, we clarify: “Therefore, in 2021, Lighthouse, I-TECH, and Medic developed a 2wT-based intervention to improve ART retention at Lighthouse’s MPC Clinic in Lilongwe, Malawi. 2wT is being tested in quasi-experimental study to determine impact on 12-month retention comparing 500 2wT participants to a historical cohort of 500 new ART initiates. As part of the overall 2wT research study, we embedded usability and acceptability testing of enrolled 2wT clients. In this process-oriented paper, we describe the HCD process for the 2wT for early ART retention app, including efforts to adapt, refine, and optimize the app for the local context. We describe the app features and functionality as a result of consistent user participation. We present mixed-methods results on client usability and acceptability with 100 early participants, 50 at 3-months and 50 at 6-months post 2wT enrollment, and report observations of participants interacting with the system.”

In discussion paragraph 3, we now added, “Usability and acceptability results suggest several areas for improvement in the 2wT client education and the system, itself. First, although clients were trained on how to respond and interact with 2wT at enrolment, there was no educational follow-up or 2wT confirmation training. Practice and repetition, complemented by refresher client education (person-led or via educational materials) during subsequent clinic visits, may be recommended as clients appear to gain confidence and competence with 2wT over time. In support of a less intensive refresher training, the 2wT team adapted the task test (Fig 3) into an educational poster with both feature and smart phone figures to promote the 2wT program and serve as a training tool to help reduce potential 2wT client orientation time.”

In the limitations, we note: “First, participants who opted into the 2wT study underwent informed consent that included both 2wT sensitization and training on responding to the 2wT system messages at enrolment. This client preparation likely contributed to usability and acceptability, but the amount is unknown. However, as 2wT orientation was only at enrollment, it is unlikely to have influenced observed improvements over time.”

2. Social desirability bias

The authors rightly identify the possibility of social desirability bias, yet they do not offer an indication of how this bias was minimized. The enhanced orientation/counselling described above may have played a role. Asymmetry of information between health care providers and research staff as well as power differential in settings like Malawi health system are important dynamics to consider. Such bias undermines the validity of the qualitative results.

Response: Thank you. As this is true for most qualitative research, especially in setting with high respect for healthcare workers and for health clinics, we did try to minimize this unknown level of bias. We added to the limitations, “Second, as with all interviewer-led questionnaires, and healthcare research in setting with high respect for HCWs, it is possible that clients wanted to please the interviewer, respond positively about 2wT, or affirm the care providers, a response bias that could have led to greater positive app feedback. However, we engaged a well-trained, highly experienced qualitative interviewer who was not involved in the 2wT program and participants were assured of total confidentiality, likely reducing this bias.”

3. Sample and sample size:

A sample was drawn purposefully and strategically; it would be good to describe the purpose and strategy that guided the sampling. In addition, among the 100 participants recruited, 28 were from a pilot. It is not clear how having participated in a pilot influences the participants familiarity of the technology and therefore, likelihood to give favorable answers on the usability 

and acceptability.

Response: We updated the entire section on study population and now added the following improved orientation: “To swiftly inform app adaptation and revision, an embedded cross-sectional, mixed methods sub-study was conducted among 100 2wT participants to assess the usability and acceptability of 2wT technology. Recruitment was done between August 2021 and January 2022. Participants were selected via a purposeful, strategic sampling approach with the aim of getting early feedback for app improvement. Therefore, we sought to enroll the first 50 participants to reach 6- and 3-months, post 2wT launch, who were willing to voluntarily participate, restricting participation to one time period. The usability participants included participants from the 2wT pilot.”

Please note that we have now filled in the questionnaire on inclusivity in global research, and a full ethics statement has been included in the methods section page 13. 

We appreciate your support, and we look forward to hearing from you.

Yours Sincerely,

Caryl Feldacker for all authors

---

## [Editor Report · Decision Letter 1]

14 Jun 2023

“It reminds me and motivates me”: usability and acceptability of an interactive, SMS-based, digital health intervention to improve early retention on antiretroviral therapy among new initiates in a high-volume, public clinic in Malawi

PONE-D-22-32318R1

Dear Dr. Feldacker,

We’re pleased to inform you that your manuscript has been judged scientifically suitable for publication and will be formally accepted for publication once it meets all outstanding technical requirements.

Kind regards,

Lisa Suzanne Dulli, PhD

Academic Editor

PLOS ONE
---

## [Editor Report · Acceptance letter]

12 Jul 2023

PONE-D-22-32318R1 

“It reminds me and motivates me”: usability and acceptability of an interactive, SMS-based, digital health intervention to improve early retention on antiretroviral therapy among new initiates in a high-volume, public clinic in Malawi 

Dear Dr. Feldacker:

I'm pleased to inform you that your manuscript has been deemed suitable for publication in PLOS ONE. Congratulations! Your manuscript is now with our production department. 

Kind regards, 

on behalf of

Dr. Lisa Suzanne Dulli 

Academic Editor

PLOS ONE